# Loss of the Novel Mitochondrial Membrane Protein FAM210B Is Associated with Hepatocellular Carcinoma

**DOI:** 10.3390/biomedicines11041232

**Published:** 2023-04-21

**Authors:** Yuanqin Zhou, Xianzhu Pan, Yakun Liu, Xiaofei Li, Keqiong Lin, Jicheng Zhu, Li Zhan, Chen Kan, Hong Zheng

**Affiliations:** 1Department of Pathophysiology, School of Basic Medicine, Anhui Medical University, Hefei 230032, China; 2Department of Pathology and Pathophysiology, School of Basic Medicine, Anhui Medical College, Hefei 230601, China

**Keywords:** hepatocellular carcinoma, mitochondrial membrane protein, FAM210B, MAPK signaling, PI3K/AKT signaling, metastasis

## Abstract

Hepatocellular carcinoma (HCC) is an aggressive and challenging disease to treat. Due to the lack of effective early diagnosis and therapy for the illness, it is crucial to identify novel biomarkers that can predict tumor behavior in HCC. In such cases, family with sequence similarity 210 member B (FAM210B) is abundant in various human tissues, but its regulatory mechanisms and role in various tissues remain unclear. In this study, we analyzed the expression pattern of FAM210B in HCC using public gene expression databases and clinical tissue samples. Our results confirmed that FAM210B was dysregulated in both HCC cell lines and HCC paraffin section samples. FAM210B depletion significantly increased the capacity of cells to grow, migrate, and invade in vitro, while overexpression of FAM210B suppressed tumor growth in a xenograft tumor model. Furthermore, we identified FAM210B’s involvement in MAPK signaling and p-AKT signaling pathways, both of which are known oncogenic signaling pathways. In summary, our study provides a rational basis for the further investigation of FAM210B as a valuable biological marker for diagnosing and predicting the prognosis of HCC patients.

## 1. Introduction

Hepatocellular carcinoma (HCC) is responsible for 85% of all liver cancer cases and is a particularly aggressive subtype of the disease [1,2]. Although surgical interventions and other medical therapies have improved survival rates for patients with early-stage HCC, the disease remains a significant threat to human health due to its high surgical risk and recurrence rate [3,4,5]. Cell hyperproliferation is the initial step in the development of HCC, which progresses into carcinoma in situ, to invasive carcinoma, and eventually to metastatic illness. However, due to the rapid growth of HCC and a lack of understanding of its mechanisms, most patients are diagnosed at an intermediate or advanced stage. Furthermore, the etiology of HCC is complex, with multiple genes playing a dynamic role at various levels [6,7]. Identifying markers that can predict tumor behavior is particularly useful for treating cancer patients because of the heterogeneity of the disease. These markers are valuable in diagnosis, staging, evaluating treatment response, detecting recurrence and distant metastases, and predicting prognosis. They also assist in improving clinical prognosis and assessment. While many biomarkers have been proposed as potential predictors of HCC progression and aggressiveness, most have not proven themselves to be practically useful [8,9]. Given that preventing and diagnosing liver cancer remains a difficult task, identifying effective markers would be of significant benefit. Moreover, by examining gene networks for changes associated with malignant behavior and progression, it may be possible to determine the fundamental mechanisms of key biomarkers that underlie HCC pathophysiology or targets for HCC therapy [10,11]. This could lead to the development of novel treatment modalities.

The FAM210B protein is relatively under-studied despite its importance. Encoded by the *FAM210B* gene located on chromosome 20, this mitochondrial inner-membrane protein is widely expressed in human tissues and plays a crucial role in hemoglobinization, proliferation, and enucleation during the final stage of erythrocyte maturation [12]. Dysfunction of *FAM210B* can negatively impact mitochondrial function and heme synthesis [13]. Additionally, FAM210B is implicated in the various biological processes of cancer cells, including cell proliferation, the inhibition of tumor cell growth, and metabolic reprogramming [14]. However, no studies have yet reported on the relationship between FAM210B and HCC.

In this study, we performed comprehensive bioinformatics analysis on the expression pattern and clinical relationship between FAM210B in HCC patients from The Cancer Genome Atlas (TCGA) [15] to uncover the function of FAM210B in HCC. Moreover, we aimed to detect the role of cell growth and migration and comprehensively discussed the underlying process in vivo and in vitro. The findings indicate that FAM210B is a tumor suppressor that may be used as a biomarker for the diagnosis of HCC and to project the prognosis of HCC patients, thereby opening new treatment avenues. 

## 2. Materials and Methods

### 2.1. Data Retrieval and Analysis

Both the TCGA and GEO databases (https://www.ncbi.nlm.nih.gov/gds/, accessed on 20 June 2022) were used for the gene expression matrix and clinical features of cohorts, including TCGA-LIHC, GSE54236 [16], GSE25097 [17] and the CPTAC confirmatory/discovery dataset (165 normal liver samples and 165 hepatocellular carcinoma samples) for HCC. The read counts were normalized and log2-transformed in both the TCGA-LIHC and CPTAC cohorts. We used a log2 transformation on the probe signal intensities for the GEO cohorts. Using SPSS 20.0, *t* tests were performed to compare the protein expression patterns between cancer and non-cancer tissues. A *p* value of <0.05 indicated a significant difference.

### 2.2. Tissue Sample Collection 

Between 2016 and 2021, HCC samples as well as adjoining non-tumor samples were obtained from the Second Affiliated Hospital of Anhui Medical University. Paraffin-embedded sections were preserved at 4 °C. All medical records included data related to patient age, sex, and other factors, including histological grade, invasion depth, tumor size, lymphatic metastasis, and TNM stage. No patients underwent any form of chemotherapy or radiation treatment preoperatively. Approval of the research was granted by the Ethics Committee of Anhui Medical University, and this study was managed in compliance with the guidance established by the Ethics Committee of Anhui Medical University. All candidates who took part in this research project provided written consent after receiving the appropriate information.

### 2.3. Survival Analysis

Protein expression profiles of FAM210B, as well as features of the patient’s survival, are outlined in Table 1 and were subjected to overall survival (OS) analysis. Subsequently, the median value of FAM210B protein expression was applied in the classification of patients into high- and low-FAM210B groups. FAM210B was correlated with OS and disease-specific survival (DSS) in liver cancer, as determined through the use of the KM Plotter [liver cancer] (http://kmplot.com, accessed on 20 June 2022). Survival statistics were analyzed using the log-rank test, with a statistical significance standard of *p* < 0.05.

### 2.4. Cell Culture

Both normal (HL02) and malignant (Hep3B, LM3, HepG2, Huh7 and PLC) human liver cell lines were used (Appendix A). High-sugar Dulbecco’s modified Eagle’s medium (DMEM; procell, Wuhan, China) with 10% fetal bovine serum (FBS, Lonsera, Canelones, Uruguay) and 1% streptomycin/penicillin was used for all cell cultures. Thereafter, the cultures were maintained at 37 °C in a sterile cell culture incubator (Memmert IF260plus, Schwabach, Germany) with humidified 5% CO_2_, and collected during their logarithmic growth phase for further use.

### 2.5. RNA Extraction and Real-Time Quantitative Reverse Transcription PCR Detection

Extraction of total RNA from cell lines was accomplished using a TRIzol kit (Invitrogen Item no.10296010, Waltham, MA, USA) and a TRIzol reagent (Invitrogen Item no.15596026) in accordance with the manufacturer’s directions. Thereafter, total RNA (2 μg) was used for cDNA synthesis based on the conditions specified: 30 min at 42 °C, 5 min at 85 °C, and storage at 4 °C. The RT reagent kit (RR037A, Takara, San Jose, CA, USA) was used for the reverse transcription of the cDNA. The SYBR green qPCR mix (RR391S, Takara, USA) was utilized for qPCR tests. Candidate lncRNA gene expression levels were normalized to 18srRNA. Three separate analyses were performed on each sample. The relative quantification of mRNAs was calculated using the 2^−ΔΔCT^ method. For qPCR experiments, the following forward (F) and reverse (R) primer sequences were utilized: FAM210B-F: 5′-TCGTTCGCTCAAGCTTGTATCTA-3′; FAM210B-R: 5′-TATGGTTGTTCACCTCTCGTTCAC-3′; GAPDH-F: 5′-AATGGCAGCAGGCACAAGTACC-3′; and GAPDH-R: 5′-CAAGGGCACAGAGACTAGCGTAATG-3′.

### 2.6. Construction of Stable Cell Lines

Lentiviral vectors were supplied by Gene Han Bio Company (Shanghai, China) for use in our study. Hep3B and HepG2 cells were infected with Lv-FAM210B or Lv-control lentiviruses in order to obtain cell lines overexpressing FAM210B. Transfection of shRNA-FAM210B (shRNA1 and 2) or control (shNC) viruses into Hep3B and HepG2 cells facilitated an analysis of the impacts of a FAM210B knockdown. Over the course of 48 h, cells were co-cultured with polybrene (2 μg/mL, Han Bio) or lentiviral vectors at an infection multiplicity of 20. Puromycin (2 μg/mL) was used to select viable cells for 14 days. Thereafter, cell cloning spheres were manually selected and re-cultured in new media. Infection efficacy was verified via qPCR assays and Western blotting experiments. The shRNA sequences that targeted *FAM210B* were shRNA1 sense: 5′-AACAGCTTGTGGATTGCATAGGCCA-3′ and antisense: 5′-TGGCCTATGCA ATCCACAAGCTGTT-3′; shRNA2 sense: 5′-TTCTCACTGGCGCAAACAGCTTG TG-3′ and antisense: 5′-CACAAGCTGTTTGCGCCAGTGAGAA-3′.

### 2.7. Cell Counting Kit-8 (CCK-8) Assay

CCK-8 (MCE, Shanghai, China) was used to detect cell proliferation capacity. Cells (2000/well) were suspended in a 96-well plate, and CCK8 levels were measured at 450 nm every 24 h over the course of 72 h per standard protocols. In each well, cells were subjected to treatment of 100 µL of DMEM and 10 µL of CCK8. All experiments were carried out in triplicate to ensure the robustness of results. The absorbance values which were recorded at each time point allowed for the generation of curves depicting the cell proliferation status.

### 2.8. Plate Cloning Experiments

After extraction, cells were rinsed three times with PBS before resuspension in a complete medium. A total of 400 total cells were placed in the 6-well plates. After the cells had been incubated for three weeks and received regular supplementation with fresh medium, they were fixed using 4% paraformaldehyde before undergoing nucleic staining using crystal violet.

### 2.9. Soft-Agar Clone-Forming Assay

The same volumes of 1.2% agar and complete media were combined, and 1.5 mL of the resulting mixture was placed into 6-well plates to generate a 0.6% base agar. A total 500 cells was resuspended in the 0.35% top agar layer. The top agar was allowed to solidify for 30 min at room temperature before being covered with 1 mL of medium. For three weeks, colonies were cultured with refreshed media which had been added to the top agar. Cell colonies were imaged using a microscope (Stemi 2000, Carl Zeiss, Jena, Germany) and manually counted.

### 2.10. Migration and Invasion Assays

We conducted Transwell Byoden chamber (pore size: 8 μm, 24-well; BD Biosciences, Franklin Lakes, NJ, USA) assays to examine cell movement. Depending on the nature of the experiment, the upper chamber was either treated with Matrigel or not as per the guidelines of the experimental protocol. In total, 4 × 10^4^ cells in a volume of 200 μL were transferred to the top chamber after collection, rinsed three times with PBS, and resuspended in serum-free media (SFM). The wells in the lower chamber were supplemented with a medium containing 10% FBS. Swabs were used to scrape the non-migrated cells from the upper filter faces. Crystal violet was used to fix the migrated cells within the lower face of the filters after 18 h. Migrated cells were then rinsed, labeled, and observed via a microscope (Axio lab.A1 microscope, Carl Zeiss, Germany). Similar inserts, also coated with Matrigel, were used for cell invasion experiments to assess the capacity of the cells for invasion.

### 2.11. Western Blot

After cell lysis in RIPA (Beyotime, Shanghai, China) with protease inhibitors (Sigma, Shanghai, China), the total protein was collected from the cells. Bicinchoninic acid (BCA) test kits were used to detect the total protein concentration. Identical quantities of proteins were separated for 90 min using SDS-PAGE at 120 V and transferred to a PVDF microfiltration membrane (Millipore, Boston, MA, USA). After washing the membrane three times with 1× Tris-based saline containing 0.05% Tween 20 (TBST), 5% bovine serum albumin was added to block the untargeted proteins at 4 °C for 2 h. The corresponding primary antibody was incubated with the membrane overnight at 4 °C. Thereafter, the samples were rinsed three times with TBST and exposed to the relevant HRP (horseradish peroxidase)-conjugated secondary antibody (8000 fold dilution, ABclonal, Wuhan, China) for two hours at 37 °C. Multiple companies, including Cell Signaling Technology (CST) (Danvers, MA, USA), Proteintech (Rosemont, IL, USA), and Novus (Centennial, CO, USA), supplied the primary antibodies used in this experiment: FAM210B (1000-fold dilution, Novus, NBP2-14523), p-ERK1/2 (1000-fold dilution, CST, 4377), ERK1/2 (1000-fold dilution, CST, 4696), p-AKT (1000-fold dilution, CST, 5012), AKT (1000-fold dilution, CST, 2920), p-p38 (1000-fold dilution, CST, 4511), p38 (1000-fold dilution, CST, 8690), and β-ACTIN (5000-fold dilution, Proteintech, HRP-60008). The secondary HRP-IgG antibodies were either goat-anti-rabbit or goat-anti-mouse and were supplied by ABclonal. Signals were detected using enhanced chemiluminescence reagents (Invigentech, Irvine, CA, USA). An ECL plus Western blotting detection system was used to visualize the proteins.

### 2.12. Immunohistochemistry (IHC) Staining

Liver cancer samples were embedded in paraffin and their protein was stained. Briefly, by utilizing xylene and a series of ethanol solutions, the samples were deparaffinized before being immersed in citrate buffer for antigen exposure. The tissues were treated with primary antibodies, including a rabbit FAM210B polyclonal antibody (1:200 dilution, Novus, NBP2-14523) and a rabbit Ki-67 polyclonal antibody (1:200 dilution, Abcam, ab15580, Cambridge, UK). The following day, HRP-labeled secondary antibodies were added to identify the target proteins, and the samples were examined with a bright-field microscope (Axio lab.A1 microscope, Carl Zeiss, Germany). Scores for the images were assigned: a score of 0 denoted negative staining, 1 denoted weak, 2 denoted medium, and 3 denoted strong staining. A final staining score ≥ 1 indicated high expression for use in statistical testing. In this section, a DAB detection kit (Brown, CTS008, R&D Systems, Minneapolis, MN, USA) was used.

### 2.13. Tumor Xenograft Model

The Nanjing Model Animal Center provided male BALB/c nude mice (4-week-old) which had been bred in pathogen-free animal care facilities. All mice were used for xenograft models. Briefly, trypsinization was performed to harvest the hepatitis virus B (HBV)-positive Hep3B cells, which were then washed three times with PBS before suspension in SFM. Hep3B cells (3 × 10^6^ cells) with FAM210B overexpression or control groups were introduced separately through subcutaneous injection into the lower back region of five nude mice in order to model HCC progression. The tumor sizes were monitored every four days with vernier calipers. After 24 days, the mice were euthanized via cervical dislocation and their neoplasms were removed for examination. All procedures involving animals in this research received prior approval from the Anhui Medical University’s Ethics Committee.

### 2.14. Statistical Analysis

GraphPad Prism8 and/or SPSS (version 22.0; SPSS Inc., Chicago, IL, USA) were used for data analysis. The link between FAM210B levels and clinicopathological variables was analyzed through a chi-square test. The KM technique and the log-rank test were employed to generate survival curves. Results from triplicate experiments were presented as mean ± SD. The statistical analysis of the variation across two groups was evaluated via Student’s *t* tests. In all in vitro tests, a one-way ANOVA was carried out to examine the variability present across the groups. A value of *p* < 0.05 denoted a significance level. Statistical significance values were set at *: *p* < 0.05; **, *p* < 0.01; ***, *p* < 0.001; ****, *p* < 0.0001.

## 3. Results

### 3.1. Patterns and Prognostic Implications of FAM210B Expression in HCC Patients

First, we conducted a search of network databases to examine variations in FAM210B expression in liver cancer and to determine its involvement in the onset and progression of HCC. We evaluated TCGA-LIHC and two GEO (GSE54236 and GSE25097) cohorts to determine the levels of *FAM210B* mRNA between HCC and normal hepatic samples. *FAM210B* levels were decreased in HCC compared to normal hepatic tissues (*p* < 0.05; Figure 1A,C,D). In addition, FAM210B protein levels were found to be lower in HCC samples compared to normal hepatic tissues from the CPTAC confirmatory/discovery dataset (Figure 1B). We further utilized the LIHC dataset derived from TCGA to confirm the correlation between FAM210B expression and HCC prognosis. The Kaplan–Meier plotter [liver cancer] showed that a low expression of FAM210B was linked with poor OS (Figure 1E) and DSS (Figure 1F) in HCC patients, as opposed to those with upregulated FAM210B. These findings demonstrate the link between low FAM210B expression and poor prognosis in patients with liver cancer.

### 3.2. The Expression Level of FAM210B Was Reduced in Clinical Tissues of HCC

To validate the observed downregulation of FAM210B in the databases, we performed immunohistochemistry (IHC) staining on 64 HCC tissues embedded in paraffin and their adjacent non-tumor tissues (Table 1). As expected, the FAM210B levels were significantly lower in tumor tissues compared to the adjacent non-tumor samples (*p* < 0.0001, Wilcoxon matched-pairs signed-ranks test; Figure 2A). Low FAM210B levels were observed in 50% (32 of 64) of HCC samples, whereas only 7.8% (5 of 64) of adjacent normal samples showed low levels of this (*p* < 0.001) (Figure 2B,C). IHC analysis also revealed that FAM210B was primarily localized in the cytoplasm of HCC tissues (Figure 2C). Subsequently, we compared the *FAM210B* mRNA expressions and protein levels in five HCC cell lines (HepG2, Huh7, Hep3B, PLC, and LM3) (Appendix A) and a hepatic immortal cell line HL02 using qPCR and Western blotting, respectively. Consistent with the tissue results, we found that *FAM210B* mRNA and protein levels were lower in hepatoma cells than in HL02 cells (Figure 2E,F).

### 3.3. Low-Expression Levels of FAM210B Are Linked to Advancement of HCC and Poor Patient Prognosis

We also investigated the association of FAM210B expression with clinicopathological parameters in HCC patients (Table 1). Using the χ^2^ test, we found a significant correlation between FAM210B levels and histological grade (*p* = 0.005), lymphatic metastasis (*p* = 0.031), and survival status (*p* = 0.021) of HCC patients (Table 1). Additionally, KM analysis was performed on HCC patients who had previously undergone hepatectomy to determine the correlation between FAM210B expression and OS. The results indicated that individuals with low FAM210B levels had a lower OS compared to those with elevated FAM210B levels (*p* = 0.002, Figure 2D). Furthermore, multivariate analysis of HCC patients indicated that FAM210B expression, lymph node metastasis, tumor size, tumor invasion, and tumor invasion depth were significant predictors of survival time (Table 2).

### 3.4. FAM210B Overexpression Inhibited HCC Cell Proliferation In Vitro

We hypothesized the involvement of FAM210B in HCC cell growth as its expression level was decreased in HCC. To investigate the functional involvement of FAM210B in HCC, we knocked down its endogenous expression in Hep3B and HepG2 cells using lentiviral shRNA vectors. We evaluated the transfection efficiency by conducting total protein and mRNA analyses, and found that FAM210B expression was suppressed by around relative to controls 70% following transfection with sh-FAM210B constructs (Appendix A). CCK-8 assays showed that the *FAM210B* knockdown enhanced cell proliferation in Hep3B and HepG2 cells (Figure 3A). Consistent with these findings, colony formation experiments demonstrated that sh-FAM210B cells generated significantly more colonies compared to shCON cells (Figure 3C). To thoroughly validate the impact of FAM210B on viability and proliferation of HCC cells, we established the overexpression of Hep3B and HepG2 cells stably FAM210B using lentivirus infection (Appendix A). Upregulation of FAM210B impaired the HepG2 and Hep3B cell reproductive capacity, as measured by CCK-8 assays and clone-forming tests (Figure 3B,D). FAM210B silencing enhanced cell tumorigenicity in Hep3B and HepG2 cells, as measured by soft agar colony formation experiments (Figure 3E), whereas FAM210B overexpression had the opposite effect (Figure 3F).

### 3.5. FAM210B Overexpression Inhibited Tumor Growth In Vivo

The overexpression of FAM210B on soft agar suppressed the anchorage-independent multiplication of HCC cells. We hypothesized that FAM210B may also influence tumor progression in vivo; we therefore established a xenograft tumor model via the subcutaneous injection of Hep3B cells stably overexpressing the FAM210B sequence (Lenti-FAM210B) or empty lentivector (Lentivector) into the dorsal flank of nude mice. Before injection, we confirmed the overexpression of FAM210B, as shown in Appendix A. After 20 days of tumor formation, we observed that the ectopic expression of FAM210B significantly suppressed tumor growth compared to the control group (Figure 4A,B). The FAM210B overexpression group had smaller tumor volumes (Figure 4B) and weights (Figure 4C). The mean tumor weight was 0.182 g in the FAM210B overexpression group and 0.45 g in the control group, resulting in a 59.56% inhibition rate (Figure 4C). Moreover, there was also no remarkable variation in total body weight across the groups (Figure 4D), indicating that cachexia was not responsible for the observed tumor growth variation. The tumor cells in the FAM210B overexpression group exhibited a nest-like infiltrative growth pattern, while those in the control group were diffuse and exhibited poor cell adhesion. Additionally, the proportion of Ki67-positive cells was significantly reduced in the FAM210B overexpression group (Figure 4E). Collectively, these findings suggest that FAM210B plays a critical role in tumor suppression in vivo.

### 3.6. FAM210B Overexpression Inhibited the Migratory and Invasive Capacities of Hepatoma Cells 

We investigated the potential effects of *FAM210B* on tumor characteristics in hepatoma cells. As FAM210B expression has been linked to tumor metastasis in HCC samples and tumor-cell metastasis and invasion are the leading causes of death associated with cancer [18], we examined the impact of FAM210B on cell migration and invasion using two representative HCC cell lines. We conducted Transwell and Matrigel chamber tests with Hep3B and HepG2 cells to investigate the effects of *FAM210B* on cell migration and invasion. Transwell chamber assays demonstrated that the loss of *FAM210B* significantly increased the migratory potential of both Hep3B and HepG2 cells compared to controls (Figure 5A,B). In contrast, FAM210B overexpression decreased cell migration (Figure 5A,B). Matrigel chamber assays revealed that loss of *FAM210B* increased the number of invading cells in both Hep3B and HepG2 cell lines, whereas *FAM210B* overexpression had the opposite effect (Figure 5C,D). Overall, our experiments indicate that FAM210B exerts a cancer-suppressing effect in HCC cell lines in vitro.

### 3.7. FAM210B Regulated ERK-AKT Signaling Pathways

We investigated whether FAM210B plays a role in intracellular signal transduction pathways, such as the mitogen-activated protein kinase (MAPK) and the phosphatidylinositol 3-kinase (PI3K)/AKT signaling pathways, which are involved in promoting metabolic processes, cell growth and survival, metastasis, and angiogenesis in response to extracellular signals [19,20]. To further elucidate FAM210B’s role in these processes and on tumor progression, we depleted FAM210B and analyzed the effect on these pathways. Western blotting analysis revealed that the loss of FAM210B substantially elevated the activity of phosphor-p38, phosphor-ERK1/2, and phosphor-AKT (Figure 6). In contrast, FAM210B overexpression significantly decreased phosphor-p38, phosphor-ERK1/2, and phosphor-AKT activity in both Hep3B and HepG2 cells (Figure 6). These findings confirm that FAM210B plays a crucial role in mediating HCC tumor suppression.

## 4. Discussion 

FAM210B appears to have roles associated with several aspects of cellular metabolism. A combination of functional abnormalities may therefore have led to the increased cell survival observed in cancer cells due to FAM210B ablation. In a previous study, FAM210B was found to be involved in erythroid differentiation and was significantly correlated with survival [12]; however, its functions within cancer cells remain poorly understood.

HCC is a complex disease with several potential contributors to its development, including viral infection [21,22,23,24], mycotoxin exposure [25], non-alcoholic fatty liver disease [26], diabetes [27], smoking [28], and gene alterations [29,30]. Therefore, it is essential to establish novel treatment approaches to improve the poor prognosis for HCC patients. In this study, we compared data from several sources, including databases and clinical samples, as well as in vivo and in vitro trials. We determined that FAM210B levels were substantially lower in HCC tissues relative to para-carcinoma tissues and in HCC cell lines as opposed to normal hepatocytes. Clinical samples indicated that low FAM210B expression levels were correlated with histological grade and lymphatic metastasis. Low expression levels were also correlated with adverse prognosis. These findings are consistent with the results in the public gene database, which indicated that lowered FAM210B expression levels were hallmarks of HCC and could be identified as prognostic factors for HCC patients. Aligned with our results is the fact that low FAM210B levels have been observed in ovarian cancer and breast cancer. The loss of FAM210B is correlated with poor OS and relapse-free survival (RFS) in ovarian and breast cancers [14]. Therefore, the role of FAM210B in HCC pathophysiology is supported and the expression of this may function as a tumor suppressor of an anti-oncogene in HCC.

Diverse tissues have distinctive responses to *FAM210B* depletion. A previous study demonstrated that the proliferation and migration abilities were markedly enhanced when *FAM210B* expression was knocked down in ovarian carcinoma SKOV3 cells and lung adenocarcinoma cancer A549 cells in vitro. Moreover, tumor formation abilities were significantly increased under this knockdown [14]. Through functional gain and loss tests in Hep3B and HepG2 cells, we determined the biological involvement of FAM210B in HCC. We discovered that lowered levels of FAM210B increase the aggression of Hep3B and HepG2 cells by enhancing their cell viability, clonality, and migratory and invasive capabilities in vitro. In contrast, FAM210B overexpression had the opposite outcomes in the same cell lines. Subsequently, the regulatory function of FAM210B on tumor progression was verified using a xenograft tumor model. These results suggested a critical function for FAM210B in the progression and metastasis of cancer. 

The PI3K/AKT pathway is known to play a central role in cell development and metabolism and is often stimulated in HCC patients, contributing to the invasiveness, metastasis, and aggressiveness of cancer cells [31,32]. AKT is a downstream signaling molecule that, when activated, phosphorylates downstream effectors involved in processes including cell proliferation, cell cycle progression, and metastasis [33,34]. Our study found that FAM210B knockdown resulted in upregulation of p-AKT, indicating activation of the PI3K/AKT pathway and suggesting that FAM210B depletion may be involved in cell growth and metastasis in HCC cells. The MAPK modules, which consisted of 3 protein kinases activated in succession, were involved in several related signal transduction pathways and implicated in cell proliferation, apoptosis, and death, and the enhancement of migratory and invasive potential of hepatoma cells [35]. Two major subgroups of the MAPK family, namely extracellular signal-regulated kinases (ERK1/2) and P38, are associated with cancer [36], and studies have shown that ERK1/2 and P38 play key roles in HCC pathogenesis [37,38,39]. In this study, we examined their levels and found that the overexpression of FAM210B substantially decreased phosphorylated ERK1/2 and P38 protein levels, while overexpression of the FAM210B knockdown resulted in elevated levels of phosphorylated ERK1/2 and P38. Our findings suggest that FAM210B is involved in the MAPK/p38 signal pathway related to cell growth and metastasis.

Our study revealed the roles of FAM210B and its effect on HCC growth, both in vivo and in vitro. However, this research has limitations due to experimental conditions, with most of the studies carried out using insufficiently comprehensive traditional experimental methods. For example, we only investigated the function of FAM210B in isolation, while the upstream and downstream pathways of FAM210B require further exploration. Therefore, the exact processes by which normal and cancerous cells react differently to FAM210B deletion/inhibition require further investigation.

## 5. Conclusions

In conclusion, our study demonstrated the dysregulation of FAM210B in both hepatoma tissues and cell lines. The ectopic expression of FAM210B resulted in a significant reduction in cell proliferation, migration, and invasion in vitro, and tumor progression was attenuated in an in vivo xenograft tumor model. Furthermore, our results indicated that FAM210B is involved in the MAPK and AKT signaling pathways, which are known to play important roles in oncogenesis. Overall, our findings suggest that FAM210B could be a valuable biomarker for diagnosing HCC and predicting patient outcomes, conclusions which warrant further investigation of this issue.

## Figures and Tables

**Figure 1 biomedicines-11-01232-f001:**
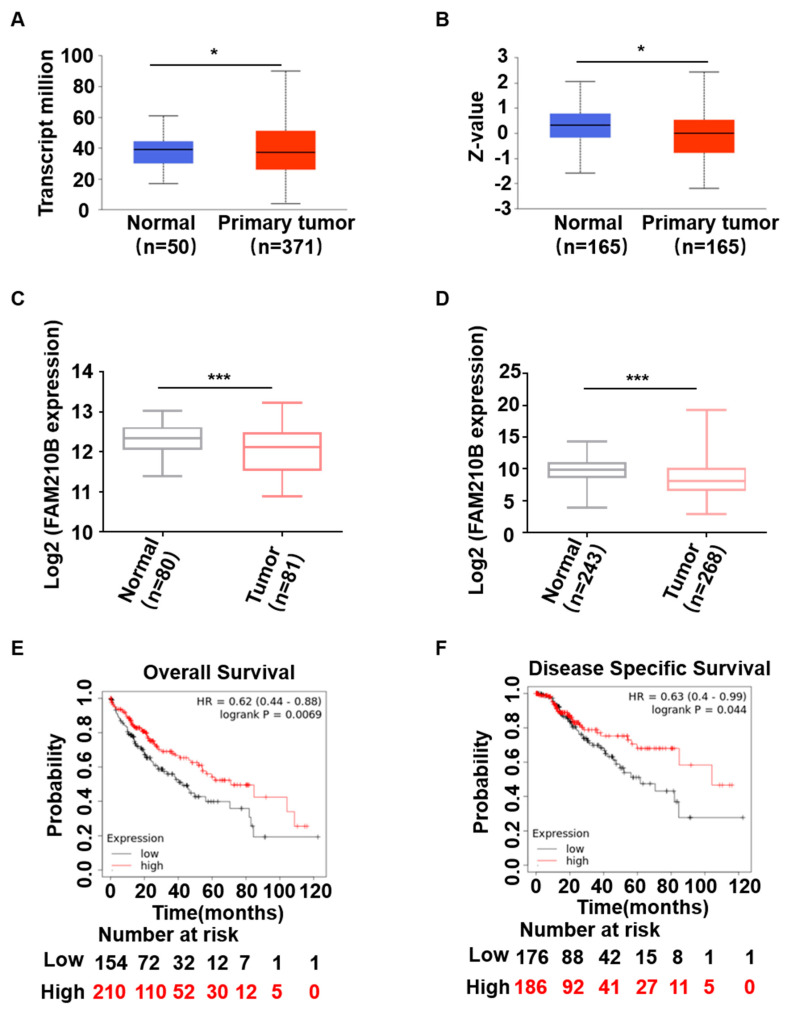
**Low expression of *FAM210B* in HCC patients and was linked to poor prognosis.** (**A**) The expression of *FAM210B* mRNA comparing the HCC tissues with the normal liver tissues in TCGA. (**B**) The expression of FAM210B protein in normal tissues or HCC samples in CPTAC confirmatory/discovery dataset. (**C**) The expression of *FAM210B* mRNA in GSE54236 normal and HCC samples. (**D**) The expression of *FAM210B* mRNA in GSE25097 adjacent non-tumor and HCC samples. (**E**,**F**) Kaplan−Meier curves showing overall survival and disease-specific survival were plotted according to the relative FAM210B expression levels from TCGA−LIHC data. *: *p* < 0.05; ***, *p* < 0.001.

**Figure 2 biomedicines-11-01232-f002:**
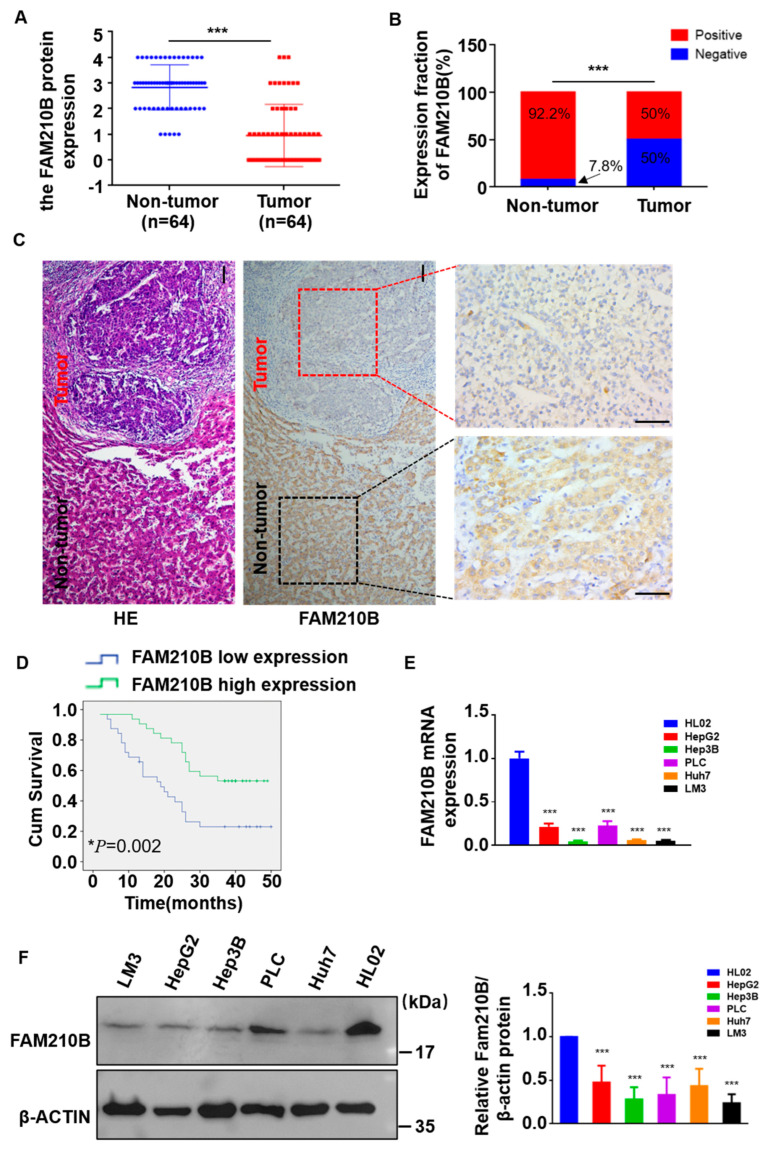
**FAM210B was frequently downregulated in HCC.** (**A**) The FAM210B levels were determined in 64 HCC and paired non−tumor paraffin samples by immunohistochemistry (IHC) assays. (**B**) The levels of FAM210B in HCC and adjoining non-tumor tissues were evaluated by IHC staining (n = 64). (**C**) IHC staining photos demonstrating FAM210B expression in HCC tumor and adjoining non-tumor samples (scale bar = 400 μm). (**D**) The overall survival of 64 individuals with HCC–related FAM210B expression was evaluated via the Kaplan–Meier method. *p* Values were derived through the log-rank test. (**E**) The levels of *FAM210B* mRNA were detected in HCC cells and in the hepatic immortal cell line HL02 by qRT−PCR assays. (**F**) The levels of FAM210B protein were detected by Western blot assays in the HCC cells and hepatic immortal cell line HL02. Quantitative analysis of Western blot is shown. *: *p* < 0.05; ***, *p* < 0.001.

**Figure 3 biomedicines-11-01232-f003:**
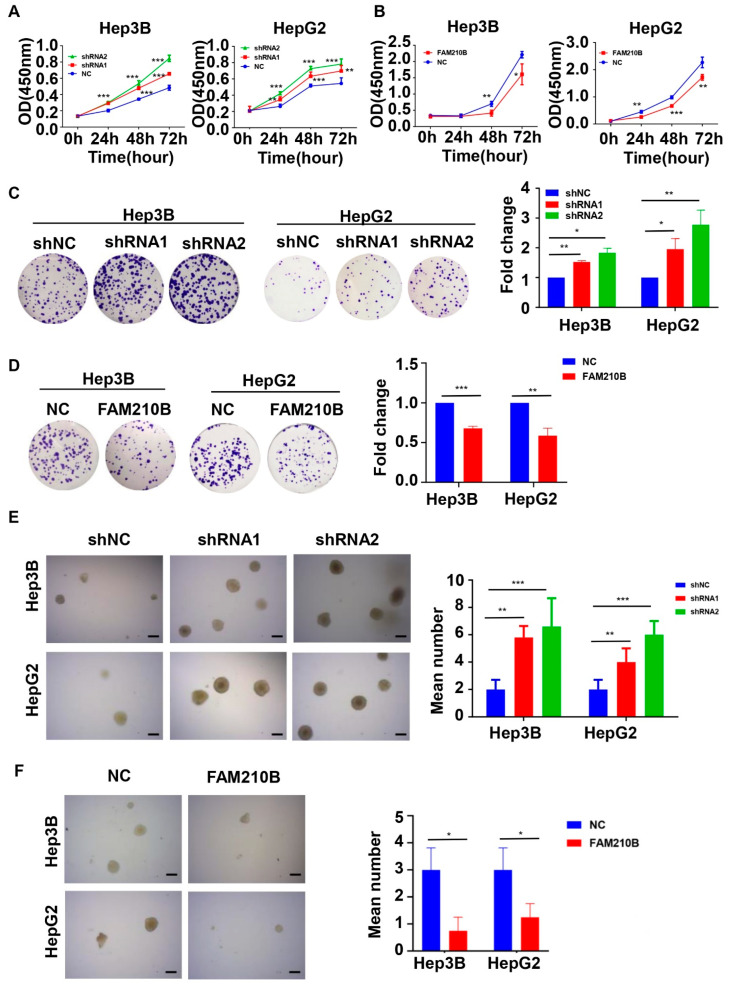
FAM210B inhibited HCC cell growth in vitro. (**A**) CCK-8 assays were used to assess the FAM210B-deficient Hep3B and HepG2 cell viability once a day for 4 days. (**B**) Overexpression of FAM210B in Hep3B and HepG2 cells inhibited cell growth, as shown by CCK-8 assays. (**C**) Clonal expansion analysis was conducted in FAM210B-deficient Hep3B and HepG2 cells. (**D**) The effects of FAM210B overexpression on the survival cells were examined using colony formation assays. When colonies were stained with crystal violet, they appeared purple. (**E**) The impacts of FAM210B knockdown on cellular malignant transformation were assessed by soft agar colony formation assay. (Scale bar = 1 mm). (**F**) The effects of FAM210B overexpression on cellular malignant transformation were assessed by soft agar colony formation assay. *: *p* < 0.05; **, *p* < 0.01; ***, *p* < 0.001.

**Figure 4 biomedicines-11-01232-f004:**
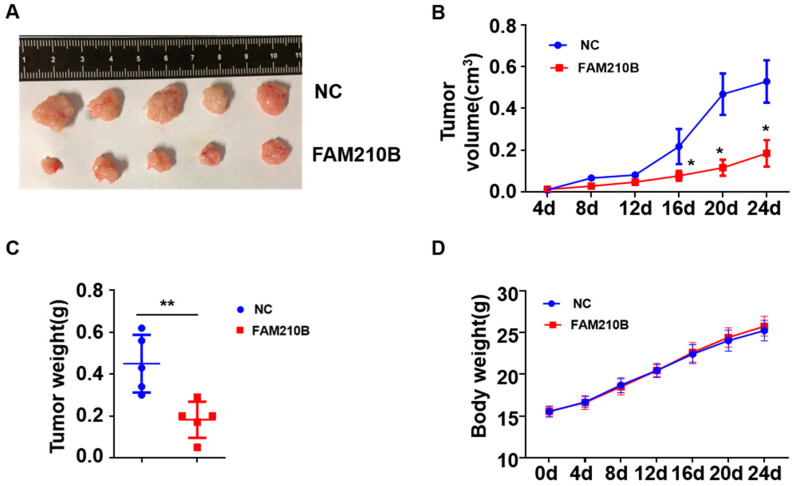
**FAM210B overexpression inhibited tumor growth in vivo.** (**A**) Xenograft tumor formation assay was performed with inoculated cell numbers of 3 × 10^6^ cells per site (n = 5/group). Images demonstrated typical tumor formation the following subcutaneous injection of FAM210B overexpressed Hep3B cells into nude mice. (**B**) The tumor volume was periodically tested for each mouse and the tumor growth curve was plotted. (**C**) The tumor mass was measured. (**D**) The body weights of mice were shown as a curve for the two groups. (**E**) FAM210B and Ki67 expression levels were detected by immunohistochemistry (Scale bar = 200 μm). *: *p* < 0.05; **, *p* < 0.01; ***, *p* < 0.001.

**Figure 5 biomedicines-11-01232-f005:**
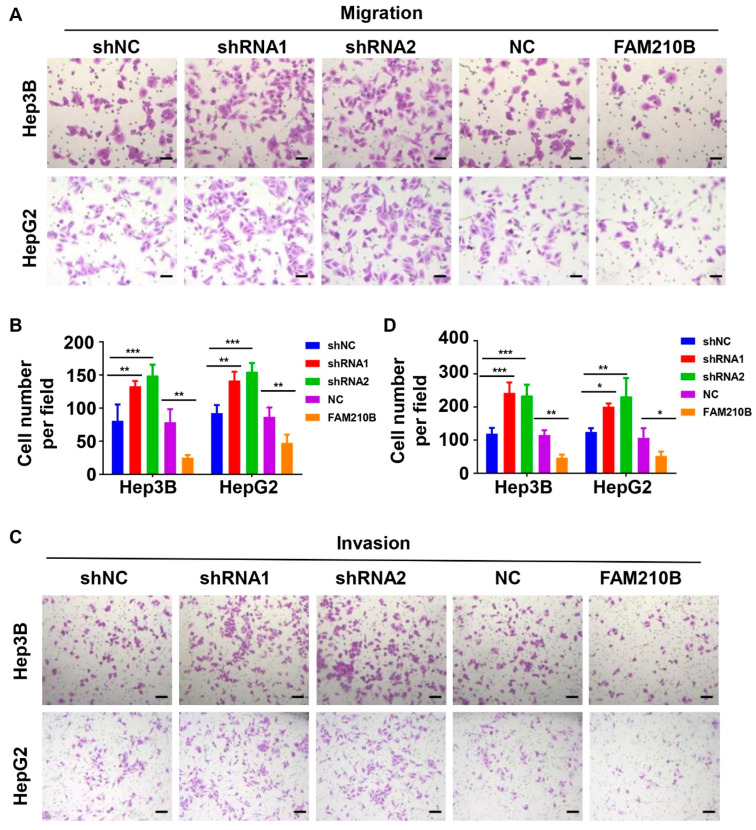
Loss of FAM210B promotes HCC cell metastasis in vitro. (**A**) The transwell migration assay was conducted in Hep3B and HepG2 cells following transfection with FAM210B shRNA lentivirus or FAM210B overexpression lentivirus. (Scale bar = 200 μm). (**B**) Statistical graphs of cell transmembrane numbers of the two groups. (**C**) Stable downregulation of FAM210B expression increased the invasive potential of Hep3B and HepG2 cells in vitro as per Matrigel chamber experiments, but upregulation of FAM210B decreased invasiveness. (Scale bar = 400 μm). (**D**) Statistical graphs of cell transmembrane number in the two groups. *: *p* < 0.05; **, *p* < 0.01; ***, *p* < 0.001.

**Figure 6 biomedicines-11-01232-f006:**
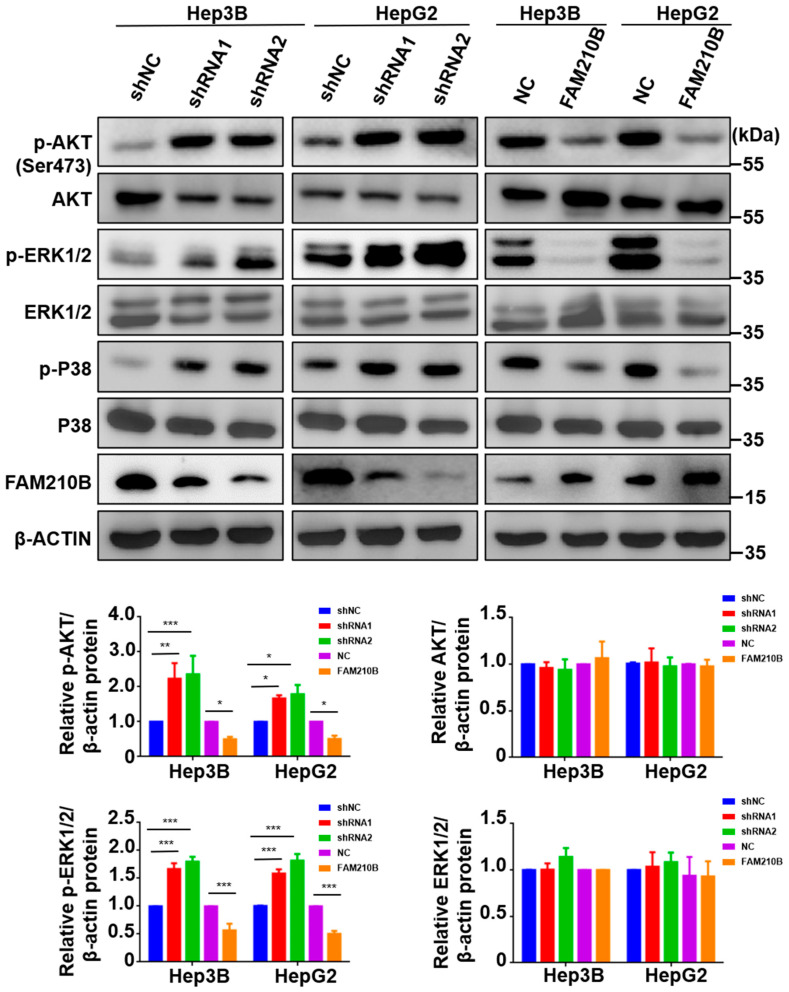
**Loss of *FAM210B* activates MAPK signaling pathway and PI3K signaling pathway.** Whole-cell lysates were generated from the HepG2 and Hep3B cells following transfection with FAM210B shRNA lentivirus or FAM210B overexpression lentivirus. Immunoblotting was examined with the indicated antibodies. The representative images and quantitative analysis were shown (n = 3). *: *p* < 0.05; **, *p* < 0.01; ***, *p* < 0.001.

**Table 1 biomedicines-11-01232-t001:** Correlation between FAM210B protein levels and clinicopathological parameters in 64 HCC patients.

Features	Total Cases	FAM210B Levels		χ^2^	^a^*p* Value
		Low	High		
Total number	64	32	32		
Gender				0.642	0.423
Male	57	27 (47.4%)	30 (52.6%)		
Female	7	5 (71.4%)	2 (28.6%)		
Age				0.567	0.451
<60	35	19 (54.3%)	16 (45.7%)		
≥60	29	13 (44.8%)	16 (55.2%)		
Tumor size (cm)				2.400	0.121
<5	40	17 (42.5%)	23 (57.5%)		
≥5	24	15 (62.5%)	9 (37.5%)		
Histological grade				- ^a^	0.005 *
High	9	3 (33.3%)	6 (66.7%)		
Middle	24	7 (29.2%)	17 (70.8%)		
Low	31	22 (71.0%)	9 (29.0%)		
Invasion depth				0.066	0.798
Tis-T1	25	12 (48.0%)	13 (52.0%)		
T2–T4	39	20 (51.3%)	19 (48.7%)		
Lymphatic metastasis				4.655	0.031 *
with	20	14 (70.0%)	6 (30.0%)		
without	44	18 (40.9%)	26 (59.1%)		
TNM stage				0.066	0.798
I–II	39	19 (48.7%)	20 (51.3%)		
III–IV	25	13 (52.0%)	12 (48.0%)		
Survival status				5.317	0.021 *
Live	25	8 (32.0%)	17 (68.0%)		
Death	39	24 (61.5%)	15 (38.5%)		

^a^ For analysis of correlation between FAM210B levels and clinicopathological parameters, Pearson’s chi-square tests were used. When the expected count of variable was less than 5, Fisher’s exact tests was used. *, *p* < 0.05.

**Table 2 biomedicines-11-01232-t002:** Univariate and multivariate analysis of overall survival in 64 patients with HCC by cox regression analysis.

Variables	Univariate Analysis		Multivariate Analysis	
	HR (95%CI)	*p* Value	HR (95%CI)	*p* Value
Gender	2.057 (0.801–5.283)	0.134	-	-
Age	0.763 (0.403–1.445)	0.407	-	-
Tumor Size (cm)	2.700 (1.435–5.082)	0.002 *	2.622 (1.164–5.908)	0.020 *
Invasion depth	5.680 (2.484–12.985)	<0.001 *	2.925 (1.033–8.283)	0.043 *
Lymphatic metastasis	15.873 (6.919–36.414)	<0.001 *	11.097 (3.693–33.346)	<0.001 *
TNM stage	6.727 (3.331–13.585)	<0.001 *	4.572 (1.763–11.856)	0.002 *
FAM210B expression	0.378 (0.197–0.726)	0.003 *	0.300 (0.125–0.722)	0.007 *

Abbreviations: *HR*, hazard ratio; *CI*, confidence interval; *, *p* < 0.05.

## Data Availability

The study’s unique contributions are detailed in this article. The raw data of the study have been deposited to the https://www.jianguoyun.com/p/DRDxesEQ6vrZChiu68gEIAA, (accessed on 20 June 2022).

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
