# Peer review of "Loss of the Novel Mitochondrial Membrane Protein FAM210B Is Associated with Hepatocellular Carcinoma"

_biomedicines, 2023, doi:10.3390/biomedicines11041232_

Round 1

Reviewer 1 Report

Hepatocellular carcinoma is known to be particularly aggressive and difficult to treat. Due to the lack of early clinical diagnosis and unsatisfactory clinical therapeutic effect, it is especially important to identify new markers that can predict tumor behavior. The authors analyzed the FAM210B expression pattern in HCC from public gene expression databases and tissue samples from clinical HCC. FAM210B was shown to be disrupted in both HCC cell lines and HCC paraffin section samples. FAM210B depletion significantly increased the ability of cells to grow, migrate, and invade in vitro, and FAM210B overexpression suppressed tumor growth in the xenograft model.

The article makes a good impression, the design is logical and consistent, the material is presented concisely and meaningfully. I think that the article will undoubtedly interest readers and should be published.

A small question: in how many parallels were studies conducted on cell lines and xenografts?

Author Response

Dear Reviewer:

Thank you very much for your comments and suggestions concerning our manuscript entitled “Loss of the novel mitochondrial membrane protein FAM210B is associated with hepatocellular carcinoma” (ID: Biomedicines-2315429). Those comments are all valuable and very helpful for revising and improving our paper, as well as the important guiding significance to our research. We have studied comments carefully and the manuscript has been revised to address the comments. Revised portions are marked by using the “Track Changes” function in the paper. It is our belief that the manuscript is substantially improved after making the suggested edits. We very much hope the revised manuscript is satisfied and accepted for publication.

Sincerely yours,

Hong Zheng on behalf of the authors

zhenghong@ahmu.edu.cn

Responses to Reviewer 1’s comments

Hepatocellular carcinoma is known to be particularly aggressive and difficult to treat. Due to the lack of early clinical diagnosis and unsatisfactory clinical therapeutic effect, it is especially important to identify new markers that can predict tumor behavior. The authors analyzed the FAM210B expression pattern in HCC from public gene expression databases and tissue samples from clinical HCC. FAM210B was shown to be disrupted in both HCC cell lines and HCC paraffin section samples. FAM210B depletion significantly increased the ability of cells to grow, migrate, and invade in vitro, and FAM210B overexpression suppressed tumor growth in the xenograft model.

The article makes a good impression, the design is logical and consistent, the material is presented concisely and meaningfully. I think that the article will undoubtedly interest readers and should be published.

A small question: in how many parallels were studies conducted on cell lines and xenografts?

Our response:

Thank you for your comments and evaluation of this manuscript. In this study, we used 3 parallels on cell lines and repeated 3 times for all experiments.  A design using 5 mice per group is made for mice xenografts experiment.

Dear Reviewer:

Thank you very much for your comments and suggestions concerning our manuscript entitled “Loss of the novel mitochondrial membrane protein FAM210B is associated with hepatocellular carcinoma” (ID: Biomedicines-2315429). Those comments are all valuable and very helpful for revising and improving our paper, as well as the important guiding significance to our research. We have studied comments carefully and the manuscript has been revised to address the comments. Revised portions are marked by using the “Track Changes” function in the paper. It is our belief that the manuscript is substantially improved after making the suggested edits. We very much hope the revised manuscript is satisfied and accepted for publication.

Responses to Reviewer 1’s comments

Hepatocellular carcinoma is known to be particularly aggressive and difficult to treat. Due to the lack of early clinical diagnosis and unsatisfactory clinical therapeutic effect, it is especially important to identify new markers that can predict tumor behavior. The authors analyzed the FAM210B expression pattern in HCC from public gene expression databases and tissue samples from clinical HCC. FAM210B was shown to be disrupted in both HCC cell lines and HCC paraffin section samples. FAM210B depletion significantly increased the ability of cells to grow, migrate, and invade in vitro, and FAM210B overexpression suppressed tumor growth in the xenograft model.

The article makes a good impression, the design is logical and consistent, the material is presented concisely and meaningfully. I think that the article will undoubtedly interest readers and should be published.

A small question: in how many parallels were studies conducted on cell lines and xenografts?

Our response:

Thank you for your comments and evaluation of this manuscript. In this study, we used 3 parallels on cell lines and repeated 3 times for all experiments.  A design using 5 mice per group is made for mice xenografts experiment.

Sincerely yours,

Hong Zheng on behalf of the authors

zhenghong@ahmu.edu.cn

Reviewer 2 Report

A study by Zhoue et al on a mitochondrial membrane protein FAM210B and its association with hepatocellular carcinoma presents a technically sound results. The manuscript is written concisely and sufficiently clearly.

Specific comments:

1. The manuscript needs editing. In addition, it is not necessary to add e.g., "in this research" as shown in Materials and Methods (2.4 Cell Culture).

2. Please transfer extensive legends from figures to figure legends (e.g., Fig. 1A and B, upper title).

3. Fig. 2A and B - please show non-tumor first.

4. Please provide quantification for all Western blots.

5. Fig. 6 - please correct ser to Ser, and indicate which ERKs do you detect (ERK1/2?)

Author Response

Dear Reviewer:

Thank you very much for your comments and suggestions concerning our manuscript entitled “Loss of the novel mitochondrial membrane protein FAM210B is associated with hepatocellular carcinoma” (ID: Biomedicines-2315429). Those comments are all valuable and very helpful for revising and improving our paper, as well as the important guiding significance to our research. We have studied comments carefully and the manuscript has been revised to address the reviewer comments. Revised portions are marked by using the “Track Changes” function in the paper. It is our belief that the manuscript is substantially improved after making the suggested edits. We very much hope the revised manuscript is satisfied and accepted for publication.

Responses to Reviewer 2’s comments

A study by Zhoue et al on a mitochondrial membrane protein FAM210B and its association with hepatocellular carcinoma presents a technically sound results. The manuscript is written concisely and sufficiently clearly.

Specific comments:

  1. The manuscript needs editing. In addition, it is not necessary to add e.g., "in this research" as shown in Materials and Methods (2.4 Cell Culture).

Our response: Thank you for your comments and evaluation of this manuscript. We have checked our manuscript carefully and made appropriate changes for some misstatements and grammatical mistakes.

  1. Please transfer extensive legends from figures to figure legends (e.g., Fig. 1A and B, upper title).

Our response: Special thanks to you for your good comments. We have made modifications in Fig.1A, B, C and D.

  1. 2A and B - please show non-tumor first.

Our response: Thank you for your suggestion. Based on you good advice, we have made modifications in Fig.2A and B.

  1. Please provide quantification for all Western blots.

Our response: Thank you for your valuable advice, we have conducted a statistical quantification of all Western blots in Fig.2F, Fig.6 and supplementary fig1f.

  1. Fig. 6 - please correct ser to Ser, and indicate which ERKs do you detect (ERK1/2?)

Our response: Thank you for your correction. Based on your suggestion, we have changed “ser” to “Ser” and “ERK” to “ERK1/2” in Fig.6.

Sincerely yours,

Hong Zheng on behalf of the authors

zhenghong@ahmu.edu.cn

Reviewer 3 Report

This manuscript, original research, performed an integrative analysis of FAM210B in Hepatocellular carcinoma.

In comparison to normal liver, HCC was characterized by lower expression. And within the HCC cases, lower expression was associated with poor prognosis of the patients. The mechanism, which was tested in vitro and in vivo, is the role of FAM210B as tumor suppressor gene.

The manuscript is well written, it is easy to read, and to understand. The results are worth publishing.

Comments:

(1) Line 72. Could you please add the manuscript references for the GSE datasets?

(2) Line 72. Could you please provide more information regarding the Clinical Proteomic Tumor Analysis Consortium (CPTAC) samples? How many subjects and other clinical data?

(3) Lines 192. Could you please change the catalog number of the primary antibody to "NBP2-14523"?

(4) Line 195. Did you really use the Stemi 2000 zeiss stereo microscope to evaluate the PPFE samples?

(5) Line 202. Could you please remind the reader writing that Hep3B cell line is an hepatitis virus B positive heatocellular carcinoma?

(6) Line 232. Instead of "dismal prognosis" I would use "unfavorable or poor prognosis"

(7) Lines 239. Could you please state that figure 1E and 1F are from TCGA-LIHC data?

(8) Regarding Table 1. The clinicopathological characteristics in the relationship with the FAM210B IHC expression is shown. Could you please indicate what dataset is being used? Is it your own primary data? How did you differentiate "low" vs "high" IHC expression?

(9) Line 251. Could you please make a table showing all the characteristics/properties of the cell lines that were used? This table could be added in the appendix.

(10) Line 251. The term "hepatoma" is being used. I think Hepatocellular carcinoma (HCC) is better.

(11) Regarding Table 2. Could you please show which dataset was used for this analysis?

(12) Lines 280-282 and Table 2. Please be careful with the hazard-ratios. The clinical variables that are associated with poor prognosis have a HR<1, and FAM210B has a HR>1. Usually, HR>1 means a poor prognosis (high associated to poor prognosis). But, this value depends on the direction of the association and the reference value. In your case, your reference is high; this is why low expression has a HR of 2.7. But this can mislead the readers. You would expect high TNM be associated with poor prognosis and higher hazards (HR>1). Would you agree?

(13) Regarding figure 4E. Could you please improve the IHC images? It is difficult to see differences.

(14) Regarding Figure 6. I can see the differences, but I wonder if you could quantify the blots. Please follow the guidelines for western blot publication of images.

(15) Have you tested the prognostic relevance of this marker in other types of tumors?

Author Response

Dear Reviewer:

Thank you very much for your comments and suggestions concerning our manuscript entitled “Loss of the novel mitochondrial membrane protein FAM210B is associated with hepatocellular carcinoma” (ID: Biomedicines-2315429). Those comments are all valuable and very helpful for revising and improving our paper, as well as the important guiding significance to our research. We have studied comments carefully and the manuscript has been revised to address the reviewer comments. Revised portions are marked by using the “Track Changes” function in the paper. It is our belief that the manuscript is substantially improved after making the suggested edits. We very much hope the revised manuscript is satisfied and accepted for publication.

Responses to Reviewer 3’s comments

This manuscript, original research, performed an integrative analysis of FAM210B in Hepatocellular carcinoma.

In comparison to normal liver, HCC was characterized by lower expression. And within the HCC cases, lower expression was associated with poor prognosis of the patients. The mechanism, which was tested in vitro and in vivo, is the role of FAM210B as tumor suppressor gene.

The manuscript is well written, it is easy to read, and to understand. The results are worth publishing.

Comments:

  • Line 72. Could you please add the manuscript references for the GSE datasets?

Our response: Thank you for your comments and evaluation of this manuscript. We have added references [16,17] for the GSE datasets in line 72(line 69 in new version).

(2) Line 72. Could you please provide more information regarding the Clinical Proteomic Tumor Analysis Consortium (CPTAC) samples? How many subjects and other clinical data?

Our Response: Thank you for your constructive suggestions. We detected the FAM210B protein expression in the CPTAC dataset for Hepatocellular carcinoma. We analyzed the expression of FAM210B protein in 165 normal liver samples and 165 hepatocellular carcinoma samples. We added this information in our manuscript (line 69 in new version).

  • Lines 192. Could you please change the catalog number of the primary antibody to "NBP2-14523"?

Our Response: Thank you for your good advice, we have made changes in line 192 and line 180 (line 174 and line 186 in new version).

  • Line 195. Did you really use the Stemi 2000 zeiss stereo microscope to evaluate the PPFE samples?

Our Response: Sorry for the mistake. We did use a Zeiss microscope to take pictures, unfortunately we confused the model (Zeiss Axio lab. A1 microscope) and we made the changes in our manuscript (line 189 in new version).

  • Line 202. Could you please remind the reader writing that Hep3B cell line is a hepatitis virus B positive heatocellular carcinoma?

Our Response: Special thanks to you for your good comments. We have corrected Hep3B cell to hepatitis virus B (HBV)-positive Hep3B cell in line 202 (line 196 in new version).

(6) Line 232. Instead of "dismal prognosis" I would use "unfavorable or poor prognosis"

Our Response: Thank you for your suggestion. We have corrected “dismal prognosis” to “poor prognosis” in line 232 (line 229 in new version).

(7) Lines 239. Could you please state that figure 1E and 1F are from TCGA-LIHC data?

Our Response: Thank you for your valuable and thoughtful comments. We have corrected in line 235 (new version).

(8) Regarding Table 1. The clinicopathological characteristics in the relationship with the FAM210B IHC expression is shown. Could you please indicate what dataset is being used? Is it your own primary data? How did you differentiate "low" vs "high" IHC expression?

Our Response: Table 1 showed our own primary data. 64 paraffin-embedded samples of HCC as well as adjoining non-tumor were gathered from the Second Affiliated Hospital of Anhui Medical University. We conducted a statistical analysis of clinicopathological characteristics in the relationship with the FAM210B IHC expression according to the patients' medical records. We scored based on two characteristics: overall staining intensity and the proportion of tissue or cells stained in IHC. The staining intensity was scored as 0 (negative), 1 (weak, light yellow), 2 (medium, yellow), and 3 (strong, deep yellow). For statistical analysis, a final staining score ≥ 1 would indicate high expression.

(9) Line 251. Could you please make a table showing all the characteristics/properties of the cell lines that were used? This table could be added in the appendix.

Our Response: Thank you for your suggestion. We have made a table showing all the characteristics/properties of the cell lines that were used in supplementary table 1.   

(10) Line 251. The term "hepatoma" is being used. I think Hepatocellular carcinoma (HCC) is better.

Our Response: Thank you for your good advice, we have changed "hepatoma" to “HCC” in our manuscript (line 245 in new version).

(11) Regarding Table 2. Could you please show which dataset was used for this analysis?

Our Response: Table 2 showed our own primary data just like Table 1. We conducted a statistical analysis of the relationship between clinical variables with prognosis.

(12) Lines 280-282 and Table 2. Please be careful with the hazard-ratios. The clinical variables that are associated with poor prognosis have a HR<1, and FAM210B has a HR>1. Usually, HR>1 means a poor prognosis (high associated to poor prognosis). But this value depends on the direction of the association and the reference value. In your case, your reference is high; this is why low expression has a HR of 2.7. But this can mislead the readers. You would expect high TNM be associated with poor prognosis and higher hazards (HR>1). Would you agree?

Our Response: Thank you for your careful attention and kind suggestion. We agreed with your points and have re-analyzed the statistical data in Table 2.

(13) Regarding figure 4E. Could you please improve the IHC images? It is difficult to see differences.

Our Response: Thank you for your comments and suggestion. We have already made a replacement in figure 4E.

(14) Regarding Figure 6. I can see the differences, but I wonder if you could quantify the blots. Please follow the guidelines for western blot publication of images.

Our Response: Thank you for your valuable advice, we have performed a statistical analysis of all Western blots in Fig.2F, Fig.6 and supplementary fig1f.

(15) Have you tested the prognostic relevance of this marker in other types of tumors?

Our Response: Thank you for your question. Currently we are conducting the related work on lung cancer and gastric cancer and needs further study.

Sincerely yours,

Hong Zheng on behalf of the authors

zhenghong@ahmu.edu.cn